# The Transcriptional Adaptor Protein ADA3a Modulates Flowering of *Arabidopsis thaliana*

**DOI:** 10.3390/cells10040904

**Published:** 2021-04-14

**Authors:** Stylianos Poulios, Despoina Dadarou, Maxim Gavriilidis, Niki Mougiou, Nestoras Kargios, Vasileia Maliori, Amy T. Hark, John H. Doonan, Konstantinos E. Vlachonasios

**Affiliations:** 1Department of Botany, Faculty of Science, School of Biology, Aristotle University of Thessaloniki, 54124 Thessaloniki, Greece; spoulios@bio.auth.gr (S.P.); D.Dadarou@warwick.ac.uk (D.D.); maxim@bio.auth.gr (M.G.); nmougiou@bio.auth.gr (N.M.); nkargios@bio.auth.gr (N.K.); maliorivasileia@gmail.com (V.M.); 2National Plant Phenomics Centre, Gogerddan Campus, Institute of Biological, Environmental, and Rural Sciences, Aberystwyth University, Aberystwyth SY23 3EE, UK; jhd2@aber.ac.uk; 3Postgraduate Program Studies “Applications of Biology—Biotechnology, Molecular and Microbial Analysis of Food and Products”, Faculty of Science, School of Biology, Aristotle University of Thessaloniki, 54124 Thessaloniki, Greece; 4Biology Department, Muhlenberg College, Allentown, PA 18104, USA; amyhark@muhlenberg.edu; 5Natural Products Research Centre of Excellence (NatPro-AUTh), Center of Interdisciplinary Research and Innovation of Aristotle University of Thessaloniki (CIRI-AUTh), 57001 Thessaloniki, Greece

**Keywords:** ADA2, ADA3, GCN5, flowering, histone acetylation, gene expression, H3K14Ac, SPL, DELLA, FT

## Abstract

Histone acetylation is directly related to gene expression. In yeast, the acetyltransferase general control nonderepressible-5 (GCN5) targets histone H3 and associates with transcriptional co-activators alteration/deficiency in activation-2 (ADA2) and alteration/deficiency in activation-3 (ADA3) in complexes like SAGA. *Arabidopsis thaliana* has two genes encoding proteins, designated ADA3a and ADA3b, that correspond to yeast ADA3. We investigated the role of ADA3a and ADA3b in regulating gene expression during flowering time. Specifically, we found that knock out mutants *ada3a-2* and the double mutant *ada3a-2 ada3b-2* lead to early flowering compared to the wild type plants under long day (LD) conditions and after moving plants from short days to LD. Consistent with ADA3a being a repressor of floral initiation, *FLOWERING LOCUS T* (*FT)* expression was increased in *ada3a* mutants. In contrast, other genes involved in multiple pathways leading to floral transition, including *FT* repressors, players in GA signaling, and members of the SPL transcriptional factors, displayed reduced expression. Chromatin immunoprecipitation analysis revealed that ADA3a affects the histone H3K14 acetylation levels in *SPL3*, *SPL5*, *RGA*, *GAI,* and *SMZ* loci. In conclusion, ADA3a is involved in floral induction through a GCN5-containing complex that acetylates histone H3 in the chromatin of flowering related genes.

## 1. Introduction

DNA packages in the nucleus of the cell in the form of chromatin. The nucleosome, the basic unit of chromatin, consists of a histone octamer, each containing two of the H2A, H2B, H3, and H4 core histones and with 147bp of DNA wrapped around it [1,2]. Chromatin is not an inert structure, but a dynamic one, constantly changing in response to external signals to regulate many DNA-related functions like replication, transcription, and DNA repair [3]. The posttranslational modification of histones governs the structure of chromatin. Multiple histone marks have been identified thus far; one of the best-studied is the acetylation of lysine residues in the N-terminal tails of core histones. Histone acetylation is affected by the opposing action of two classes of enzymes: histone acetyltransferases (HATs) and histone deacetylases (HDACs) [4]. The transfer of the acetyl-group to the lysine neutralizes its positive charge and thus weakens the interaction between histone and DNA, relaxing the structure of chromatin [3]. HATs are often found as part of larger multiprotein complexes, like the Spt–Ada–Gcn5–acetyltransferase (SAGA) complex [5]. Other proteins within these complexes play essential roles in controlling enzyme recruitment, activity, and substrate specificity [3].

The first report about the biological role of alteration/deficiency in Activation-3 (Ada3) genes was in yeast. The gene encodes for a transcriptional adaptor protein, bridging transcriptional activators with the basal transcription machinery [6,7,8,9,10]. Yeast *ada3* mutants are viable but grow slowly in minimal media and are temperature sensitive [6,7,8,9,10]. The yeast Ada3 protein interacts with alteration/deficiency in Activation-2 (Ada2) and general control nonderepressible-5 (Gcn5) proteins forming a trimeric complex [7,9,11]. Further biochemical characterization revealed that Ada3, along with Ada2 and Gcn5, is part of larger multiprotein complexes like SAGA and ADA in yeast [5,12,13]. These complexes have multiple functions, acting as transcriptional adaptors and histone acetyltransferases. These functions indicate the significance of histone acetylation during the initial steps of transcriptional activation, mediated by interactions between transcriptional activators and the general transcription machinery [5]. Ada3 protein has been identified in evolutionarily diverse organisms, including *Drosophila melanogaster* [14], *Mus musculus* [15], and *Homo sapiens* [8]. In Drosophila, ADA3 is necessary for cell viability and growth; *ada3* mutants are all lethal in the pupal stage [16]. Loss of function of ADA3 in the germline of *Mus musculus* is embryonically lethal, and cell deletion of ADA3 leads to abnormal cell cycle progression, indicating that ADA3 protein is critical both at the organismic and cellular level in mammals [15].

The *Arabidopsis thaliana* genome encodes several components of SAGA complex, including: one gene for the *GCN5* histone acetyltransferase; two homologues of yeast Ada2 transcriptional adaptor, named *ADA2a* and *ADA2b* [17]; and two SAGA-associated factor 29 (*SGF29)* genes [18]. Previous work identified one homologue of yeast and human ADA3 in *Arabidopsis thaliana*: the gene At4g29790 [19,20]. However, a recent review suggested that the Arabidopsis genome encodes for two ADA3 homologues [21]. Mutations in *ADA2b* and *GCN5* have severe developmental defects including dwarfism, impaired meristem development, sterility or sub-fertility, aberrant root development, and altered abiotic and biotic stress responses [21,22,23,24,25,26,27,28,29,30,31]. These results suggest that both genes are involved in a plethora of different biological processes. On the contrary, *ada2a* mutant plants display a wild-type phenotype, and overexpression of ADA2a was unable to complement the loss of *ADA2b* function [32], indicating that the two proteins have distinct biochemical roles. Furthermore, mutations in Arabidopsis *SGF29* genes do not display obvious developmental problems and could have an auxiliary role in salt stress responses [18].

The transition from the vegetative phase to reproductive development, commonly referred to as floral transition or flowering, is a critical process for the survival of plant species [33]. It is controlled at multiple levels by genetically defined pathways, ensuring that the plant flowers when the environmental, as well as the internal, conditions are appropriate. In *Arabidopsis thaliana*, six routes were defined including: (a) the photoperiod pathway, referring to the regulation of flowering in response to the length of the day and the quality of the perceived light; (b) the vernalization pathway, which regulates flowering in response to exposure to a long period of cold; (c) the gibberellin (GA) pathway, which refers to the need for GA for normal flowering; (d) the autonomous path, which refers to endogenous regulators that work independently from the other routes; (e) a pathway influenced by plant age; and (f) pathways that allow modulation by ambient temperature [34,35,36,37].

The objective of this study was to identify homologues of yeast ADA3 in *Arabidopsis thaliana* and explore their biological role. We found that there are two homologs in the Arabidopsis genome and that only ADA3a has a clear role in flowering initiation, affecting histone acetylation and the gene expression profile across several pathways.

## 2. Materials and Methods

### 2.1. Plant Material and Growth Conditions 

The Arabidopsis thaliana (L) Heynh. mutants *ada3a-1* (SALK_032897), *ada3a-2* (SALK_055660), *ada3b-1* (SALK_035588), and *ada3b-2* (SALK_042026), all in Columbia-0 background, were obtained from Arabidopsis Biological Research Centre (ABRC). The *ada2a-3* mutant was previously described [32]. We created the *ada3a-2ada3b-2* double mutant by using pollen from *ada3a-2* to fertilize *ada3b-2* gynoecium. The triple mutant *ada2a-3ada3a-2ada3b-2* was obtained using pollen from *ada2a-3* stamens to fertilize the gynoecium of mature *ada3a-2ada3b-2* flowers. The F1 plants self-fertilized, and the double or triple mutants were identified in the F2 and F3 using PCR-based genotyping (Appendix A). The physical location of ADA2a, ADA3a, and ADA3b in the Arabidopsis genome is shown on Appendix A. 

Seeds were surface-sterilized with a commercially available bleach solution (30% in water) and stratified at 4 °C for 3–4 days in the dark. For plating, seeds were sown on Gamborg B5 medium at pH 5.7 (Ducheffa, Amsterdam, Netherlands), supplemented with 1% sucrose (Ducheffa) and 0.8% phytoagar (Ducheffa). Plants were grown at 20–22 °C with 100–150 μmol m^2^ s^−1^ cool-white fluorescent lamps under long-day conditions (16 h light/8 h dark). Seven to ten day-old plants were transferred to soil to complete their development. The commercially available soil, Terrahum^®^ (Deutsche Kompost Handelsgesellschaft, Geeste, Germany) was used for cultivation. Soil-grown plants were irrigated twice weekly with water.

### 2.2. Flowering Experiments

For flowering time experiments, seeds were stratified for three days at 4 °C in the dark and sown directly on the soil. Plants were grown at 20–22 °C with 100–150 μmol m^2^ s^−1^ cool-white fluorescent lamps under long-day (LD) conditions or short-days for 30 days and shifted to LD. The plants were monitored daily to record the day of meristem emergence and the number of rosette leaves. Rosette leaves were counted when the inflorescence was approximately 0.5–1 cm. Photos were taken with a Panasonic Lumix camera.

### 2.3. Gene Expression Analysis

The expression of *ADA3a* and *ADA3b* genes recorded in rosette leaves from 3-week-old plants of Col-0 and *ada3a-1*, *ada3a-2*, *ada3b-1*, and *ada3b-2* mutants. For the analysis of flowering genes, the aboveground tissues of 10-day-old plants, grown under long-day conditions, were collected at the end of the 16th hour of light (Zeitgeber Time 16-ZT16) and flash-frozen in liquid nitrogen. RNA extraction was performed using a Nucleospin^®^ RNA plant kit (Macherey-Nagel, Duren, Germany). Reverse was transcription carried out using 0.5 μg total RNA with a PrimeScript^TM^ 1st strand cDNA Synthesis Kit (TaKaRa, Shiga, Japan) in three or four independent biological repeats. Quantitative reverse-transcription polymerase chain reaction (RT-qPCR) analysis was performed using a KAPA^TM^ SYBR^®^ Green Fast qPCR kit (Kapa Biosystems, Cape Town, South Africa) or AMPLIFYME SG Universal Mix (Blirt, Gdańsk, Poland) and the ABI StepOne^TM^ platform (Applied Biosystems, Foster City, CA, USA). For the calculation of expression levels, the ΔΔCt method was applied, according to the manufacturer’s instructions (Applied Biosystems, Foster City, CA, USA). The Ct values obtained from the genes were normalized to the values obtained from *PDF2* or *At4G26410* [38]. The values presented in the results are expressed as *PDF2*-normalized levels of the target genes. Student’s t-test was used to compare the expression of the target genes between wild-type (WT) and mutant plants to determine whether the difference was significant at *p* ≤ 0.05, *p* ≤ 0.01, or *p* ≤ 0.001.

### 2.4. Chromatin-Immunoprecipitation (ChIP)-qPCR

For ChIP, the aboveground tissues of 10-day-old seedlings grown on soil under long-days at ZT16 were harvested in 1xPBS. The tissues were fixed with 1% formaldehyde in PBS for 15 min under vacuum and treated with 0.125 M glycine for 5 min under vacuum to end the cross-linking reaction. The tissue was washed twice with ice-cold 1xPBS solution, dried in paper, and weighted (approximately 300 mg of tissue was used for the immunoprecipitation from each sample). Tissue was ground to a fine powder in liquid nitrogen and ChIP assays were performed according to [39], with minor modifications. The isolated chromatin was sheared to an average length of 500 bp by sonication five times for 10 s each. The sonicated chromatin was diluted ten times, and 1mL used for each immunoprecipitation. We used antibodies against acetylated histone H3K14 (Anti-Histone H3 (Lys14), EMD Millipore #07-353) and H3 (ChIPAb + Histone H3 C-term, EMD Millipore #17-10046). The immunoprecipitated samples were incubated for 60 min with protein A–agarose beads (Cell Signaling, Danvers, MA, USA). The chromatin fragments attached to the beads were eluted at 65 °C with 1% SDS, 0.1 M NaHCO3. Proteins were detached from the chromatin by reverse cross-linking with 200 mM NaCl at 65 °C overnight, followed by proteinase K (Sigma-Aldrich, St. Louis, MO, USA) treatment. The DNA was isolated using a commercially available PCR clean-up kit (Macherey-Nagel, Duren, Germany). Immunoprecipitated DNA was diluted in water and analyzed by qPCR using specific primers (Appendix A). Real-time PCR was carried out in reactions using a buffer containing AmplifyMe SG Universal Mix (Blirt, Gdańsk, Poland), using the StepOne^TM^ platform (Applied Biosystems, Foster City, CA, USA). Input samples were used in five 10-fold serial dilutions to construct a standard curve. All data obtained by q-PCR were presented as a percentage of input. The immunoprecipitated sample values were normalized to the input. The ratio of H3K14 to H3 values is shown in the figures. Student’s t-test was used to compare acetylation levels between wild-type and mutant plants to determine whether the difference was significant at *p* ≤ 0.05, *p* ≤ 0.01, or *p* ≤ 0.001.

### 2.5. Phylogenetic Analysis

We used the ADA3 domain from yeast ADA3 protein to blast the *Arabidopsis thaliana* genome for putative ADA3 domain-containing proteins. Then the ADA3 domain from the two Arabidopsis homologues was aligned to yeast and *Kazachstania africana* ADA3 protein using clustal omega (EMBL-EBI) and visualized with ESPript3.0 [40]. Furthermore, the ADA3 domain from other plant species’ ADA3 orthologues was compared with the ADA3 domain from *Danio rerio*, *Gallus gallus*, *Mus musculus*, *Homo sapiens,* and *Drosophila melanogaster*. The phylogenetic analysis and the evolutionary history of ADA3 protein from Brassicaceae was inferred with the MEGA X [41] program using neighbor-joining [42] with the nucleotide substitutional model K80 and the node support was assessed using 1000 bootstrap replicates. The evolutionary distances were computed using the Poisson correction method [43] and are in units of the number of amino acid substitutions per site. The rate variation among sites was modelled with a gamma distribution (shape parameter = 1). All positions with less than 50% site coverage were eliminated. The prediction of the secondary structure of the ADA3a and ADA3b protein was made using JPred4 [44].

### 2.6. Yeast Two Hybrid Assay

Yeast two hybrid (Y2H) assays were performed using the Matchmaker Gold System (Takara, Shiga, Japan). cDNA fragments encoding putative members of the plant SAGA complex, ADA3a, ADA3b, SGF29a, SGF29b, TAF12b, and SGF11, were PCR amplified from the pda04096 (RIKEN), U14407, U11423, U82343, U11077, and U12870 plasmids obtained by ABRC. Primers were designed using the SnapGene^®^ software under default parameters (Appendix A). PCR amplification was performed with high fidelity Q5 polymerase (New England Biolabs). All the plasmids used in the Y2H were constructed using a Gibson Assembly^®^ kit (New England Biolabs, Ipswich, MA, USA), according to the manufacturer’s instruction. Both vectors (pGBKT7 and pGADT7) were cut by NdeI and BamHI (New England Biolabs, Ipswich, MA, USA) to create linear plasmid with sticky ends. The constructed plasmids were transformed into *E. coli* cells, which were plated on LB Agar plates with the appropriate antibiotic depending on the plasmid. Plasmids were then isolated using a NucleoSpin^®^ Plasmid kit (Macherey-Nagel, Duren, Germany). GCN5, ADA2a, and ADA2b in pGBKT7 and pGADT7 were described by [26]. The yeast strain AH109 was transformed with pGBKT7 containing studied bait genes, while the Y187 was transformed with pGADT7 vector containing the prey genes of interest. Yeast cultures were grown in low stringency medium and diluted to an optical density at 600 nm (OD600) of 1.0. Tenfold serial dilutions (10-fold, 100-fold, and 1000-fold) were made in sterile H_2_O, and 10 μL of each was plated on high stringency media and allow to grow for 5 days at 30 °C. Yeast cells carrying pGBKT7 empty vectors were used as negative controls, while yeast cells carrying ADA2a-AD and GCN5-BD, as well as ADA2b-AD and GCN5-BD fusion proteins, were used as positive controls [26]. 

## 3. Results

### 3.1. The Arabidopsis Thaliana Genome Encodes Two ADA3 Homologues

To identify homologues of yeast ADA3 in the *Arabidopsis thaliana* genome, we used the yeast Ada3 protein sequence as a query to search in protein sequences from the plant kingdom. Two potential candidates were observed, named *ADA3a* (*At2g19390*) and *ADA3b* (*At4g29790*). Previous works identified only *At4g29790* as the yeast and human ADA3 homologue [19,20]. These proteins include a domain that resembles the ADA3 domain of other organisms (Figure 1A). The ADA3 domain from several eukaryotic organisms was used to construct a phylogenetic tree. The tree separates the ADA3 plant group into two smaller subgroups, demonstrating that, contrary to other eukaryotes, in *Arabidopsis thaliana* two genes encode proteins that resemble yeast ADA3 (Figure 1B). The deduced AtADA3a and AtADA3b polypeptides consist of 1211 amino acids and have predicted molecular weights of 133.36 and 133.24 kDa, respectively, and share 75% amino acid sequence identity. The predicted secondary structure of the two proteins indicates that the N terminus of ADA3 proteins has a conserved coil–coil domain followed by a nuclear localization signal. In the center of the protein, from approximately 580 through 720 aa, a putative a/b hydrolase domain was detected followed by the ADA3 domain, from 743–970 aa. The ADA3 domain was predicted to have six b-strands and five a-helices, with a putative DNA and RNA binding domain. Finally, the C-terminus is conserved and expected to have a b-strand structure (Appendix A). The two genes that encoded putative ADA3 proteins also exist in other plants, including relatives of Arabidopsis (Appendix A). Phylogenetic analysis of Viridiplantae putative ADA3 proteins revealed that ADA3 is absent only in chlorophytes. The first appearance of ADA3 protein was observed in charophytes, such as *Klebsormidium nitens* and *Chara braunii,* followed by liverworts, mosses, and lycophytes, ANA clade, monocots, and eudicots (Figure 2) [21]. Searching for motifs in ADA3 polypeptide sequences in plant models, including Arabidopsis, Brachypodium, Amborella, Selaginella, Physcomitrella, and Marchantia indicated that the primary structure of the ADA3 proteins is highly conserved (Appendix A). Both ADA3a and ADA3b are expressed in most organs and tissues, but ADA3b has higher levels of expression than ADA3a (Appendix A). The subcellular localization of ADA3a was predicted to be nuclear, whereas for ADA3b the prediction was nuclear and cytoplasmic (Appendix A).

### 3.2. Molecular and Phenotypic Characterization of T-DNA Mutants of the ADA3a and ADA3b in Arabidopsis thaliana

To elucidate the biological role of *ADA3a* and *ADA3b* in *Arabidopsis thaliana*, we screened for plants bearing T-DNA disruption mutations in *ADA3a* and *ADA3b*. Homozygous T-DNA mutants of both ADA3 genes were characterized and designated as *ada3a-1* (SALK_032897), *ada3a-2* (SALK_055660), *ada3b-1* (SALK_035588), and *ada3b-2* (SALK_042026). The T-DNAs are located in the fourth and ninth exons of *ADA3a* and the first and eighth exons of *ADA3b*, respectively (Figure 3A). Gene expression analysis by RT-PCR in rosette leaves was used to assess the effect of T-DNA insertion site on the *ADA3a* and *ADA3b* transcripts in homozygous mutant plants. In all homozygous mutants tested, gene expression analysis failed to detect a full-length transcript product (Figure 3B,C), indicating that T-DNA insertion affects the integrity of *ADA3a* and *ADA3b* transcripts, respectively. Therefore, we concluded that the expression of full-length *ADA3a* and *ADA3b* transcripts are greatly diminished in the *ada3a* or *ada3b* mutants, and that all mutants represent loss-of-function alleles. Furthermore, the *ADA3a* expression level was not significantly affected in *ada3b-2*, and the *ADA3b* expression was not significantly altered in *ada3a-2* mutant plants compared to wild-type plants (Figure 3D). Moreover, both mutant plants did not significantly alter the expression of ADA2a, ADA2b, and GCN5 genes relative to wild-type plants (Appendix A). The phenotype of homozygous mutants *ada3a-1*, *ada3a-2*, *ada3b-1*, and *ada3b-2* was examined, especially for characteristics previously altered in *gcn5* and *ada2b* mutant plants [22]. All the homozygous mutants displayed normal leaf growth and development (Figure 4), suggesting that *ADA3a* and *ADA3b* genes are not critical for the vegetative stage growth of *Arabidopsis thaliana*. Then we explored if there is functional redundancy between *ADA3a* and *ADA3b* genes. The phenotype of the *ada3a-2ada3b-2* double mutant was normal, suggesting that *ADA3a* and *ADA3b* have no significant role in *Arabidopsis thaliana* vegetative growth.

### 3.3. ADA3a Transcriptional Adaptor Is a Negative Regulator of Flowering in Arabidopsis thaliana

Time-resolved phenotypic characterization revealed that *ada3a* mutants had an early flowering phenotype in comparison to wild-type under long-day conditions (Figure 4; Table 1 and Appendix A; Appendix A). Specifically, *ada3a-1* and *ada3a-2* mutants had a reduced number of rosette leaves and flowered earlier compared to wild type and *ada3b* mutant plants (Table 1 and Appendix A). These differences were statistically significant, and the experiments were repeated at least six times with similar results. Therefore, we concluded that *ada3a* mutants flower earlier than the wild-type plants, suggesting that ADA3a is involved in the transition from the vegetative to the reproductive stage of Arabidopsis. To check if ADA3b has a redundant role in flowering initiation, we monitored the flowering behavior of the *ada3a-2ada3b-2* double mutant. We observed that the double mutant also displayed an early flowering phenotype similar to *ada3a*, suggesting that ADA3a is a negative regulator of flowering in *Arabidopsis thaliana* and that ADA3b is not involved in flowering under permissive growth conditions. Then we examined the flowering behavior of plants grown under short-day conditions for 30 days and then moved to long-day conditions. The *ada3a-2* mutant plants had lower numbers of rosette leaves, the inflorescence emergence was visible four days earlier, and the first flower opened five days earlier than the wild-type plants (Table 2). The inflorescence of *ada3b-2* mutant plants emerged two days earlier, and the first flower opened three days earlier than the wild type plants. However, the number of rosette leaves was not significantly lower than the wild type plants. Finally, the double *ada3a-2ada3b-2* mutant plants also displayed an early flowering phenotype, similarly to *ada3a-2* single mutants (Table 2). These results suggest that ADA3a modulated flowering when plants moved from SD to LD conditions, with ADA3b playing a more modest role.

### 3.4. ADA3a and ADA3b Genetically Interact with ADA2a to Affect Flowering Time in Arabidopsis thaliana

To test if ADA3a and ADA3b interact genetically with other HAT module components of GCN5-related complexes in the regulation of flowering time, we examined the role of ADA2a (At3g07740), since both ADA2b and GCN5 mutants cause severe defects in vegetative growth [32]. The *ada2a-3* mutant plants had fewer rosette leaves under long-day conditions [32]; therefore, we created the *ada2a-3ada3a-2* and *ada2a-3ada3b-2* double mutants, as well as the *ada2a-3ada3a-2ada3b-2* triple mutant. The *ada2a-3ada3a-2*, *ada2a-3ada3b-2* double, and the *ada2a-3ada3a-2ada3b-2* triple mutants displayed a similar phenotype to wild-type plants, without any synergistic effect on vegetative growth (Figure 5). Interestingly, the *ada2a-3* mutation enhanced the early flowering phenotype of *ada3a-2ada3b-2,* as shown by the number of rosette leaves and days to bolting, but did not enhance the flowering phenotype of the single *ada3a-2* mutants (Table 3) significantly. Furthermore, the *ada2a-3ada3b-2* double mutant had fewer rosette leaves than *ada3b-2* and caused earlier emergence of the inflorescence meristem than the single *ada2a-3* and *ada3b-2* mutants (Table 3; Figure 5). These results suggest an additive genetic interaction between ADA2a, ADA3a, and ADA3b during flowering initiation.

### 3.5. ADA3a and ADA3b Transcriptional Adaptors Regulate Gene Expression in Multiple Flowering Pathways

To explore the molecular mechanisms by which ADA3a affects flowering, we monitored the expression of several flowering-related genes [34] by RT-qPCR (Figure 6). The flowering promoting gene *FT* was highly overexpressed in *ada3a-2* and *ada3a-2ada3b-2* mutants but not in the *ada3b-2* mutant, confirming the early flowering phenotype and suggesting that ADA3a is a negative regulator of *FT* gene expression (Figure 6A). The gene expression levels of the negative regulator *FLC* and positive regulator *SOC1* were similar in the mutants and the wild type plants, indicating that the ADA3a and ADA3b did not affect their levels (Figure 6B,C). SQUAMOSA promoter binding protein-like (SPL) genes are involved in vegetative phase transitions and regulated by the Arabidopsis SAGA-like complex [45]. To test if *SPLs* gene expression is also affected by ADA3a and ADA3b, we analyzed the gene expression of *SPL3*, *SPL5,* and *SPL9* (Figure 6D–F). The expression of *SPL3* and *SPL5* was significantly reduced in *ada3a-2* and *ada3a-2ada3b-2* in comparison to wild type and *ada3b-2* single mutant plants (Figure 6D,E). This result indicates that ADA3a affects *SPL3* and *SPL5* expression in a positive manner. Notably, the expression levels of *SPL9* were not affected by mutations in ADA3a and ADA3b (Figure 6F). Next, we analyzed the expression of two DELLA genes that are involved in GA signaling and affect flowering in Arabidopsis. Both *RGA* and *GAI* were downregulated in *ada3a-2* and *ada3a-2ada3b-2* (Figure 6G,H), suggesting that ADA3a is a positive regulator of RGA and GAI gene expression. The *FT* locus receives multiple signals from different pathways [34]. To further investigate how the gene expression of *FT* was affected, we analyzed the expression of negative regulators of FT in *ada3a* and *ada3b* mutant plants. The expression of CYCLING DOF FACTOR 1 (CDF1), a gene that suppresses the expression of CONSTANS (CO) and FT [46], was significantly reduced in *ada3a-2* and *ada3a-2ada3b-2* (Figure 6I). Other FT repressors include two transcription factors of the RAV family; TEMPRANILLO1 (TEM1) and TEMPRANILLO2 (TEM2) and four genes of the euAP2 lineage; SCHLAFMUTZE (SMZ), SNARCHZAPFEN (SNZ), TARGET OF EAT1 (TOE1), and TOE2 [34]. The *TEM1* and *TEM2* genes had no change in their expression in the *ada3a* and *ada3b* mutant plants (Appendix A), while the expression of *TOE1* and *TOE2* was significantly downregulated in the *ada3a-2ada3b-2* double and *ada3a-2* single mutants, respectively (Appendix A). Interestingly, the expression of *SMZ* and *SNZ* was reduced in both *ada3a-2* and *ada3b-2* single mutants, and *SMZ* was also downregulated in *ada3a-2ada3b-2* (Figure 6J,K). This result indicates that the gene expression of several FT repressors is affected by ADA3a and ADA3b function. Finally, the expression of the *FCA* gene from the autonomous pathway [47] was significantly reduced in both single and the double mutants (Figure 6L) indicating that *FCA* could also be a target of ADA3a and ADA3b. Overall our data suggested that mainly ADA3a affects the expression of many flowering related genes from different pathways.

### 3.6. ADA3a Affect Histone Acetylation Levels in SPL3 and SPL5 as Well as Genes from Multiple Flowering Pathways, Including SMZ, RGA, and GAI Genes

To examine whether the observed changes in gene expression in the ada3a-2 mutants are correlated with the histone acetylation status of their locus, we performed ChIP analysis using antibodies for histone H3 lysine 14 acetylation normalized to H3. We found that in the proximal promoter regions (region 1), before the transcriptional start site of SPL3 and SPL5 locus, histone H3K14 acetylation was reduced significantly in ada3a-2 mutants in comparison to wild-type plants (Figure 7A,B). In region 2, located in the first exon, a statistically significant reduction of H3K14 acetylation was detected in ada3a-2 mutants only for SPL3 gene (Figure 7A,B). Furthermore, in the 3′UTR of SPL3 and SPL5, the H3K14 acetylation levels were reduced in ada3a-2 mutants, but not statistically significant. These results suggest that ADA3a is required for H3K14 acetylation in the SPL3 and SPL5 loci. Then we investigated the effect of ada3a-2 mutant plants in histone acetylation in SMZ locus. Two regions in the SMZ locus were examined, the first in the proximal promoter (1) and the second in the ORF (2). In both areas, H3K14 acetylation was reduced in ada3a-2 mutant plants, with the promoter region being statistically significant. This result suggests that ADA3a is required for proper H3K14 acetylation in the SMZ locus (Figure 7C). Then we investigated the effect of ADA3a on the acetylation levels of proximal promoter regions of the CDF1 gene. The H3K14 acetylation levels in the promoter of CDF1 were not affected by ada3a-2 mutant, suggesting that the reduction in the CDF1 expression was not a result of impaired H3K14 acetylation (Figure 7D). Interestingly in both promoters of RGA and GAI genes, the H3K14 acetylation levels were reduced (Figure 7E,F), suggesting that ADA3a affects gene expression by altering the H3K14 acetylation levels in their promoters.

### 3.7. ADA3a and ADA3b Interact with Members of the Histone Acetylation Module (HAT) of the Arabidopsis thaliana SAGA-Like Complex

To identify potential protein–protein interactions between ADA3a and ADA3b proteins and other members of the SAGA complex we performed yeast two hybrid assays. We tested the pairwise interaction of ADA3a with ADA3b, SGF29a, SGF29b, GCN5, ADA2b, and SGF11, and the pairwise interaction of ADA3b with SGF29a, SGF29b, ADA2b, and TAF12b. We observed that ADA3a and ADA3b interacted, suggesting that they could form and act as a heterodimer (Figure 8). Furthermore, ADA3b interacts with both SGF29a and SGF29b. However in our system no interactions were detected with other HAT module like ADA2b, GCN5, or components of other SAGA modules, like SGF11 and TAF12b (Figure 8). These results suggest that ADA3a and ADA3b interact with other members of the HAT module of a SAGA-like complex in Arabidopsis thaliana.

## 4. Discussion

In this study, we explored the biological role of the Arabidopsis homologues of yeast Ada3 transcriptional adaptor. In comparison to yeast and other eukaryotes, *Arabidopsis thaliana* has two ADA3 genes, designated ADA3a and ADA3b. Other members of the yeast SAGA complex also have two orthologues in *Arabidopsis thaliana*, like ADA2a and ADA2b, as well as SGF29a and SGF29b [21]. ADA3 protein is vital for the development of other eukaryotes [8,15,16]. In contrast, the Arabidopsis ADA3 proteins are not essential for the completion of the biological cycle, and the mutants exhibited no easily observable developmental phenotypes like *gcn5* and *ada2b* mutants [22]. Our results indicate that the peripheral members of the SAGA HAT module [48] do not strongly affect plant development, because the activity of GCN5 and the associated partner ADA2b is not impaired by mutations in ADA2a, ADA3, and SGF29 genes. Using yeast two hybrid assays we revealed that ADA3a interacts with ADA3b, whereas ADA3b could interact with SGF29a and SGF29b, suggesting that both ADA3 proteins could be part of several GCN5-containing complex in plants. The presence of ADA3b, but not ADA3a, was identified as part of the SAGA components in Arabidopsis suspension culture cells [49,50]. Further biochemical approaches are required to unveil the interactions of both ADA3 proteins with other components of SAGA complex in planta. 

Our results suggest that ADA3a is a negative regulator of flowering, in plants grown under LD conditions or during the transition from SD to LD conditions. ADA3a affects the expression of FT, a central floral integrator [33] but this interaction could be indirect, since several negative regulators of FT like SMZ, SNZ, and CDF1 [51,52,53] are also downregulated in *ada3a-2* and/or *ada3a-2ada3b-2* mutant plants. Furthermore, *gcn5* and *ada2b* mutants do not directly affect *FT* expression [40]. Overexpression of *SMZ* and *SNZ* caused a late-flowering phenotype under LD conditions [53]. Moreover, SMZ protein acts as a repressor of flowering by binding directly to the *FT* locus and down-regulating its expression [53]. Our data suggest that histone H3K14 acetylation in the SMZ locus was affected by ADA3a. Further studies are required to identify biochemical interactions between GCN5 containing complex and binding factors in the SMZ locus. Another factor that affects the floral transition is CDF1, which is known to repress the expression of floral activator genes such as CO and FT in Arabidopsis [51]. Our results suggest that the expression of CDF1 is affected by ADA3a, but contrary to SMZ, is not correlated with alteration in histone H3K14 acetylation, implying that this effect is not, at least directly, regulated by GCN5-containing complex. 

Gibberellins promote flowering, primarily in plants grown under LD conditions [54]. The bioactive GAs bind to their receptors and trigger the degradation of DELLA proteins. DELLA proteins are expressed in companion cells of leaf phloem and delay flowering by reducing the expression of FT under LD conditions [55]. Our results suggest that the expression of the DELLA genes, RGA, and GAI was affected by ADA3a, and this is correlated with reduced H3K14 acetylation levels in their promoters, suggesting a direct effect on these targets. In mature rosette leaves of *gcn5* and *ada2b* mutants, a decreased expression of *GAI* was also observed [22,51]. Therefore, the GCN5-containing complex that contains ADA3a is required for the proper expression of RGA and GAI proteins and the effect of GA on flowering initiation. 

Plant age is also an essential factor in controlling flowering initiation. Several SPL transcription factors are associated with the age-dependent floral transition, including SPL3, 4, 5, 9, and 15 [56]. Our results indicate that ADA3a affects the expression of *SPL3* and *SPL5.* These factors are regulated by GCN5-ADA2b-containing complexes and contribute to the transition from juvenile to adult phases [45]. As a result, GCN5 and ADA2b promote flowering by determining the transcription of SPLs, a mechanism that is essential for the miR156-independent induction of SPLs [45]. The promotion of flowering by SPL3/4/5 is closely associated with FT [57]. Furthermore, SPL3 is known to bind directly to GTAC motif in the FT promoter, controlling the flowering time in response to ambient temperature [58]. Moreover, GA and DELLA proteins regulate flowering partially through direct interaction with SPL transcription factors [59,60,61]. Therefore, our results suggest that ADA3a could serve as part of the GCN5-containing complex to properly acetylate H3K14 in the locus of *SPL3* and *SPL5*. Subsequently, ADA3a may accelerate SPLs dependent phase transitions by environmental factors, including flowering [45]. ADA3b is not strongly involved in flowering under permissive conditions, since neither the single nor the double mutants flowered earlier than *ada3a* mutants. Previous studies showed that ADA2a had a minor effect on flowering initiation under long-day conditions [32]. Under our conditions, we showed that the *ada2a-3* enhanced flowering phenotypes of *ada3a-2ada3b-2* and *ada3b-2*, suggesting that ADA2a, ADA3a, and ADA3b interact genetically to control flowering time. On the other hand, mutation of *gcn5* delayed flowering without affecting FT levels [45]. The difference found between the flowering habits of *gcn5* and *ada3a* mutants arises from the multiple developmental processes that GCN5 is involved in. In Arabidopsis, ADA3a and ADA3b could act by enhancing the GCN5 action or interacting with transcription factors, as observed in yeast [13]. The reduced expression of *SPL3,* a known GCN5 target [45], and its reduced H3K14 acetylation in *ada3a* mutants suggests that GCN5-dependent histone acetylation could be compromised, at least partially. However, the expression levels of GCN5 and ADA2b were not affected by ADA3a and ADA3b, suggesting that the HAT module is functional in *ada3* mutants. As a result, the role of ADA3a in plant development and flowering is auxiliary and could potentially uncover a hidden role of the SAGA complex in mechanisms of flower initiation. Our results suggest that a putative GCN5-containing complex modulates several factors involved in the floral transition, possibly through the ADA3a protein. Further studies are required to understand fully the biochemical interactions between ADA3a and other components of the SAGA complex with transcriptional factors that affect the flowering transition to elucidate the mechanistic role that histone acetylation plays during the transition from vegetative to the reproductive stage in Arabidopsis.

## 5. Conclusions

Taken together, ADA3a, a likely component of the SAGA complex in Arabidopsis, affects multiple floral pathways such as age, gibberellin signaling, and *FT* repressors. 

## Figures and Tables

**Figure 1 cells-10-00904-f001:**
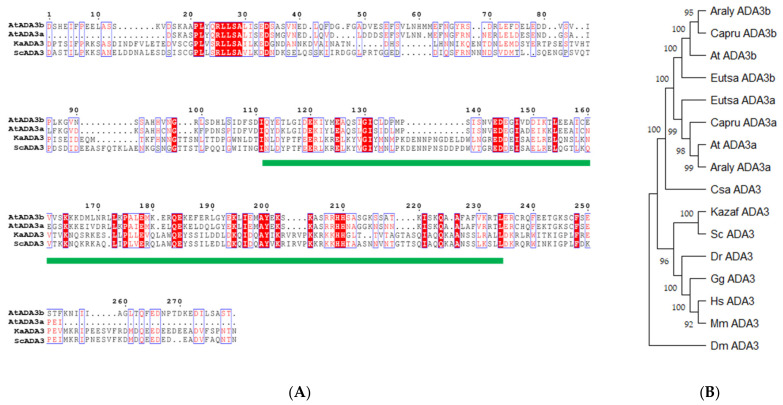
Alignment of ADA3 domain (**A**) and phylogenetic tree for ADA3 domain across organisms (**B**). The ADA3 domain is highlighted with a green line.

**Figure 2 cells-10-00904-f002:**
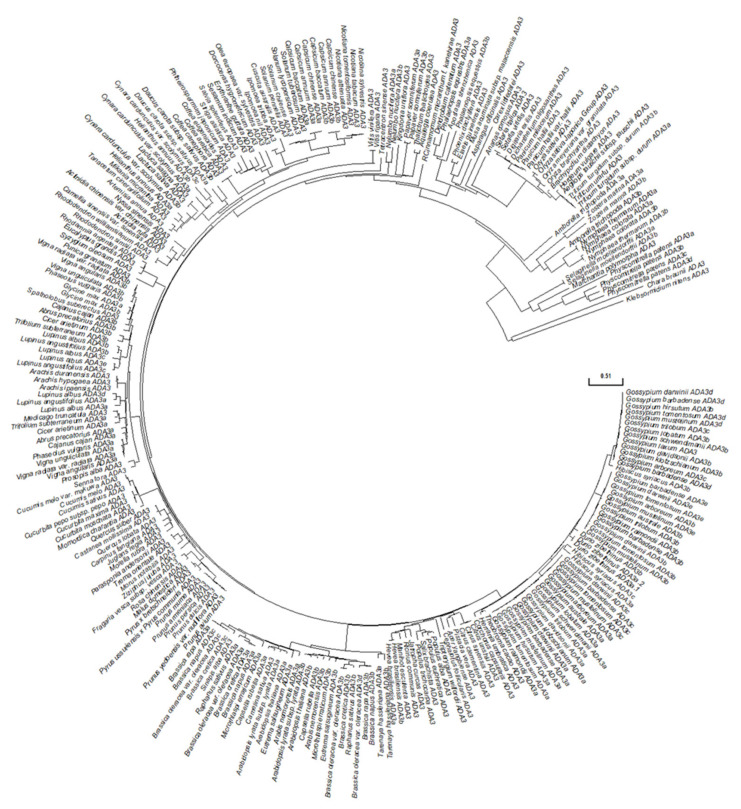
Phylogenetic analysis of ADA3 proteins in Viridiplantae.

**Figure 3 cells-10-00904-f003:**
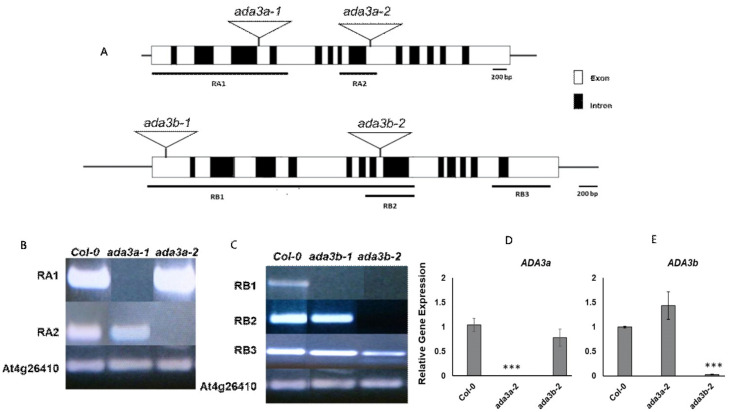
Schematic diagram of the Arabidopsis ADA3 genes (**A**), with ADA3a shown at the top and ADA3b shown below. Exons are denoted as white boxes, and triangular schemes indicate the position of the T-DNA insertion creating each mutant allele. The numbered horizontal bars below the ADA3a and ADA3b genes schematic refer to the coding sequence assayed in (**B**) and (**C**). Reverse-transcriptase (RT)-PCR of ADA3a and ADA3b transcripts (**B**,**C**). Plant genotypes are indicated at the top of each column, and the amplified fragments of the ADA3a transcript (**B**) or ADA3b transcript (**C**) are shown to the left of each row. RT-PCR of At4g26410 were used as controls for cDNA amount and quality. Real-time RT-qPCR analysis of *ADA3a* (**D**) and *ADA3b* (**E**) transcripts in wild type Col-0, *ada3a-2* and *ada3b-2* 10 days-old, soil-grown plants, under long day conditions. Statistical significance calculated by *t*-test, *** *p* < 0.001.

**Figure 4 cells-10-00904-f004:**
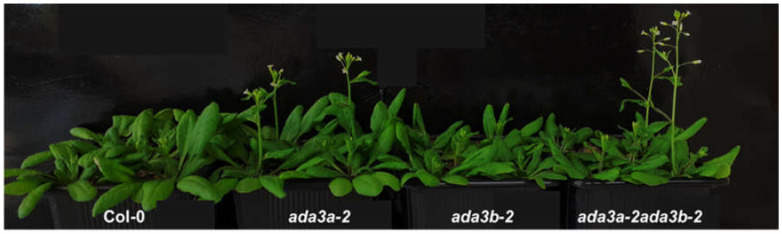
ADA3a affected flowering in Arabidopsis. Photo of approximately 4-week-old Col-0, *ada3a-2*, *ada3b-2*, and *ada3a-2ada3b-2* mutant plants grown under long-day conditions.

**Figure 5 cells-10-00904-f005:**
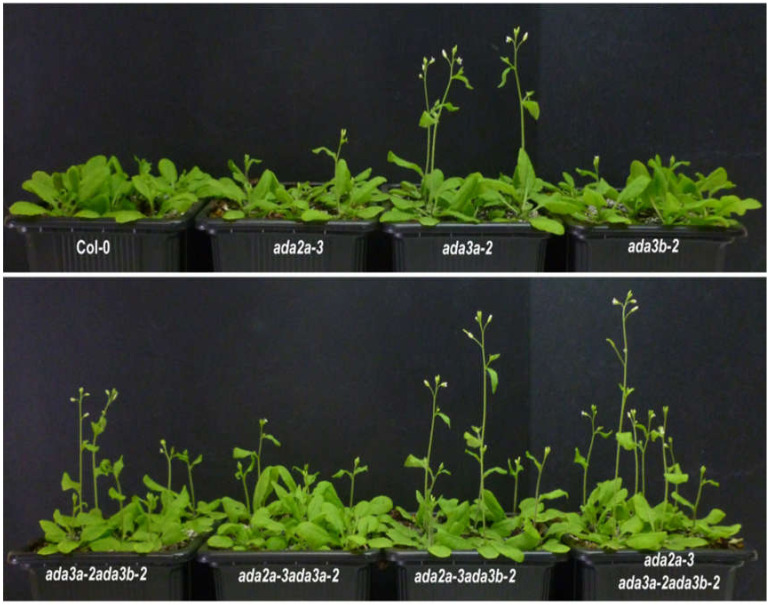
The effect of ADA2a, ADA3a, and ADA3b on flowering. Four-week-old Col-0, *ada2a-3*, *ada3a-2*, *ada3b-2*, *ada3a-2ada3b-2*, *ada2a-3ada3a-2*, *ada2a-3ada3b-2,* and *ada2a-3ada3a-2ada3b-2* mutant plants grown under long-day conditions.

**Figure 6 cells-10-00904-f006:**
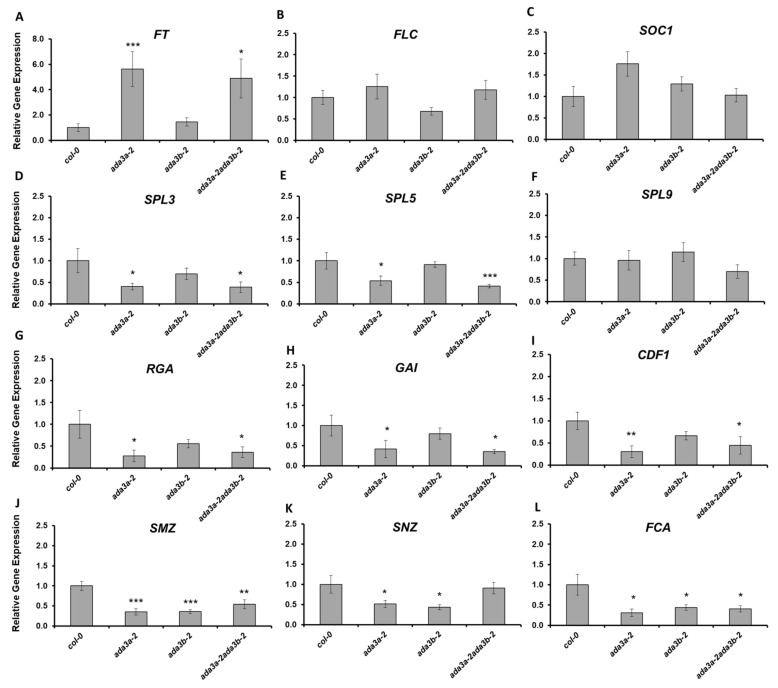
The effect of ADA3a and ADA3b on the flowering related gene expression under long-day conditions. Gene expression analysis of *FT* (**A**), *FLC* (**B**), *SOC1* (**C**), *SPL3* (**D**), *SPL5* (**E**), SPL9 (**F**), RGA (**G**), GAI (**H**), CDF1 (**I**), SMZ (**J**), SNZ (**K**), and FCA (**L**) in the wild-type, *ada3a-2*, *ada3b-2,* and *ada3a-2ada3b-2* plants, quantified by reverse transcriptase (RT)-qPCR. Statistical significance calculated by t-test, * *p* < 0.05, ** *p* < 0.01, *** *p* < 0.001.

**Figure 7 cells-10-00904-f007:**
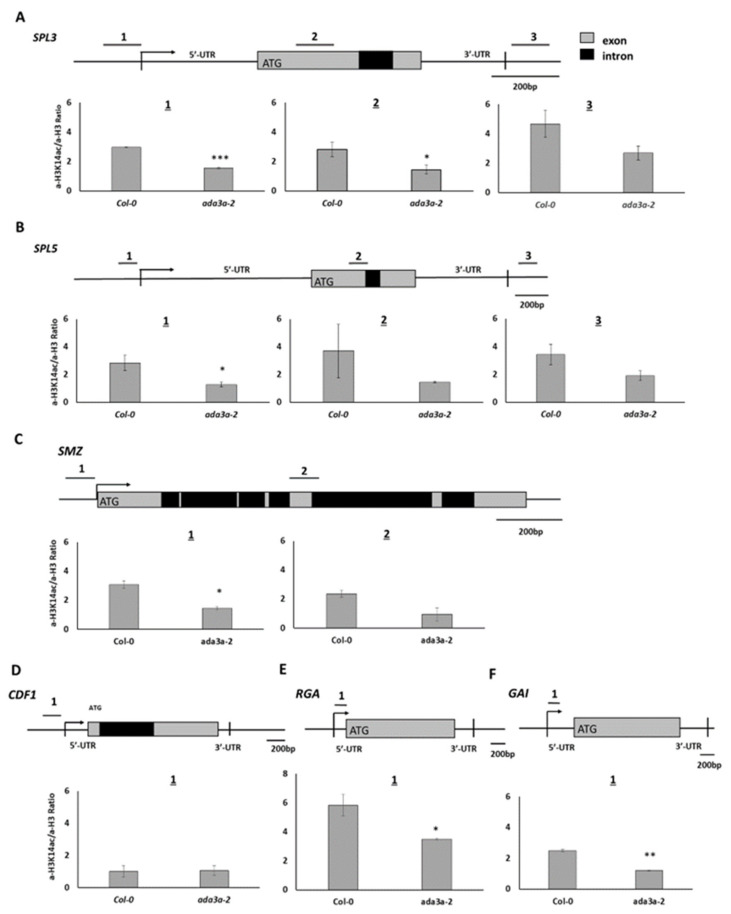
The acetylation status of H3K14 in the *Arabidopsis thaliana* Col-0 and *ada3a-2* ten-days old plants. (**A**) Three genomic regions for the *SPL3* and (**B**) *SPL5* locus were analyzed: near the transcription start site (1), in the protein-coding region (2), and after the 3′ untranslated region (3). (**C**) Two genomic regions were analyzed for the *SMZ* locus: one near the transcription start site (1), and one in the protein-coding region (2). (**D**) One genomic region was analyzed for the *CDF1* locus, near the transcription start site, (**E**,**F**) for the *RGA* and *GAI* locus, within the 5′ untranslated regions. Values were obtained by q-PCR as a percentage of input. Antibody for histone H3 was used as an internal control, and the ratio (percentage of input with H3K14ac/percentage of input with H3) is presented. The error bars represent standard error (SE) of three technical repeats. Statistical significance was calculated by *t*-test, * *p* < 0.05, ** *p* < 0.01, *** *p* < 0.001.

**Figure 8 cells-10-00904-f008:**
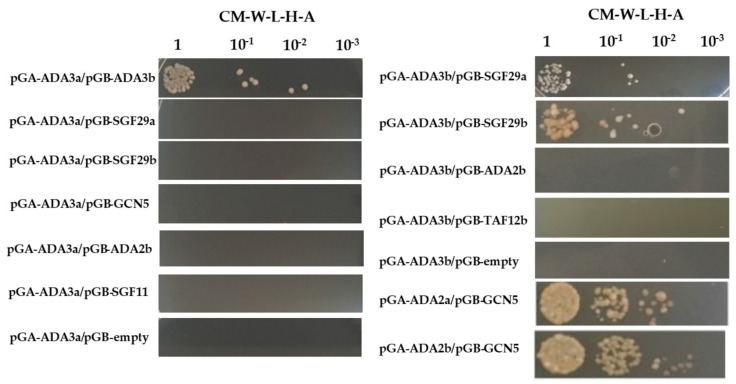
Yeast two hybrid assays of protein-protein interactions between ADA3a and ADA3b and other members of the *Arabidopsis thaliana* SAGA-like complex.

**Table 1 cells-10-00904-t001:** ADA3a is involved in flowering. Leaf number and days to bolting in ada3a and ada3b single and double mutant plants.

Genotypes	Number of Leaves ^1^	*p*-Value	Days to Bolting ^2^	*p*-Value	N
Col-0	8.38 ± 0.90		21.54 ± 1.07		26
*ada3a-2*	7.50 ± 0.99	0.0015	20.38 ± 1.10	0.0003	26
*ada3b-2*	8.33 ± 0.84	0.8306	21.07 ± 1.17	0.1247	30
*ada3a-2ada3b-2*	7.46 ± 1.14	0.0019	20.39 ± 1.26	0.0007	28

^1^ Leaf number counted when the inflorescence length exceeded 0.5 cm. Means and standard deviation. ^2^ Days from germination to inflorescence stem approximately 0.5 cm. Statistical significance calculated by t-test, *p*-value <0.05, *p*-value <0.01, *p*-value <0.001.

**Table 2 cells-10-00904-t002:** ADA3a and ADA3b are involved in the vegetative to reproductive transition. Plants were moved from SD to LD, and the number of leaves, days to bolting, and days to first open flower were counted.

Genotypes	Number of Leaves ^1^	*p*-Value	Days to Bolting ^2^	*p*-Value	Days to 1st Open Flower ^3^	*p*-Value	N
Col-0	28.1 ± 1.92		14.6 ± 0.86		19.2 ± 0.83		17
*ada3a-2*	24.5 ± 2.98	0.0002	10.2 ± 1.68	<0.0001	14.1 ± 1.98	<0.0001	17
*ada3b-2*	26.4 ± 3.02	0.0668	12.0 ± 2.98	0.0013	15.9 ± 3.38	0.0005	17
*ada3a-2ada3b-2*	24.0 ± 2.67	0.0001	10.9 ± 2.85	<0.0001	15.0 ± 3.61	<0.0001	15

^1^ Rosette leaf number counted when the inflorescence length exceeded 0.5 cm. ^2^ Days from the transfer from short-day (SD) to long-day (LD) conditions until the inflorescence length exceeded 0.5 cm. ^3^ Days when the first opened flower observed. Statistical significance calculated by *t*-test, *p*-value <0.05, *p*-value < 0.01, *p*-value < 0.001.

**Table 3 cells-10-00904-t003:** The effect of *ADA2a* on the flowering phenotype of *ada3* mutants. Leaf number and days to bolting in single, double, and triple mutant plants.

Genotypes	Number of Leaves ^1^	*p*-Value	Days to Bolting ^2^	*p*-Value	N
Col-0	8.50 ± 1.20		20.77 ± 1.59		30
*ada2a-3*	8.33 ± 0.85	0.7487	20.45 ± 1.77	0.4638	33
*ada3a-2*	7.71 ± 0.99	0.0060	18.97 ± 2.28	0.0004	35
*ada3b-2*	9.15 ± 1.01	0.0306	20.62 ± 2.35	0.7836	18
*ada3a-2ada3b-2*	7.70 ± 0.98	0.0054	19.06 ± 1.46	<0.0001	33
*ada2a-3ada3a-2*	7.82 ± 0.72	0.0097	19.44 ± 1.35	0.0007	34
*ada2a-3ada3b-2*	8.28 ± 0.77	0.3998	18.63 ± 1.39	<0.0001	32
*ada2a-3ada3a-2ada3b-2*	7.47 ± 0.76	0.0002	18.19 ± 1.47	<0.0001	32

^1^ Leaf number counted when the inflorescence length exceeded 0.5 cm. Means and standard deviation. ^2^ Days from germination to inflorescence stem approximately 0.5 cm. Statistical significance calculated by *t*-test, *p*-value < 0.05, *p*-value < 0.01, *p*-value < 0.001.

## Data Availability

The data supporting the findings of this study are available from the corresponding author, (Konstantinos Vlachonasios), upon request.

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
