# Peer review of "The Transcriptional Adaptor Protein ADA3a Modulates Flowering of Arabidopsis thaliana"

_cells, 2021, doi:10.3390/cells10040904_

Round 1

Reviewer 1 Report

The manuscript described the transcriptional adaptor protein ADA3a modulates flowering of Arabidopsis thaliana. The experimental results are systematic and complete. Some minor suggestions are giving as following:

  1. In Figure 1b, The names of species can be abbreviated.
  2. In discussion, why GCN5 and ADA3 have different role in plant development should be discussed.
  3. I suggestion the author maybe can provide the results of protein-protein interaction about ADA3 and others,such as, ADA2,GCN5.

Author Response

We would like to thank the reviewer for the positive comments. In the revised manuscript we address all the request.

Req.1  In Figure 1b, The names of species can be abbreviated.

Response 1: We have abbreviated the names of the species in a new Fig. 1b.

Req. 2. In discussion, why GCN5 and ADA3 have different role in plant development should be discussed.

Response 2: We rewrote the discussion part (ln 516-524) by providing some explanation on the different role of GCN5 and ADA3a. We also provide data as a supplementary figure S7 and explain these findings on ln 262-263 to support the discussion.

Req 3. I suggestion the author maybe can provide the results of protein-protein interaction about ADA3 and others,such as, ADA2,GCN5.

Response 3:  We provide some preliminary data on protein-protein interaction of ADA3 with other components of the SAGA complex based on yeast two-hybrid assays. We added a new section on Materials and Methods (ln 188-209), a new figure 8 and a new session in results (3.7, ln 423-434) and a paragraph in discussion (ln 461-467).

Reviewer 2 Report

Flowering is a trait controlled by a complex genetic system interacting with environmental conditions. Understanding genes affecting floral transition has significant impacts on both basic and applied sciences. This manuscript is of great value for scientific literature. It is well written except some minor deficiencies that need authors’ revision for the following.

Abstract:  Indicate what the FT and FT stand for. These had not been mentioned in Introduction section, so the readers will be confused not knowing if it is abbreviation for flowering time or floral transition. Thus, authors should provide the full terms of these abbreviations in front of the parentheses containing them.

Introduction: Is there a full term for ADA and other abbreviated names?

Results: Line 203 – change “found” to “existed” in other plants.

Table 3: If the bolded p-values for number of leaves were used to indicate significance of difference, this practice should also be used for all p-values in all tables.

Suggestions:

Authors could provide the relative physical location of ADA2 and ADA3 in Arabidopsis thaliana based on the starting and ending base-pair figures in relation to the full length of scaffolds in its genome assembly. This information will be interesting to some researchers because flowering genes are essential for seed-producing plants.

Author Response

We would like the reviewer for the suggestions and the posivite comments in our manuscript. In the revised manuscript we have addressed all the requests.

Specifically

Req 1. Abstract:  Indicate what the FT and FT stand for. These had not been mentioned in Introduction section, so the readers will be confused not knowing if it is abbreviation for flowering time or floral transition. Thus, authors should provide the full terms of these abbreviations in front of the parentheses containing them.

Introduction: Is there a full term for ADA and other abbreviated names?

Response 1

We have provided all abbreviation requested, FT, ADA2, ADA3, GCN5, SGF29 etc, both in the abstract as well as in the introduction and results.

Request 2

Results: Line 203 – change “found” to “existed” in other plants.

Response 2:  please see the change in ln 230.

Request 3

Table 3: If the bolded p-values for number of leaves were used to indicate significance of difference, this practice should also be used for all p-values in all tables.

Response 3

We thank the reviewer for spot it this. We add a phrase in every table to indicate what the p-value means. "Statistical significance calculated by t-test, *P-value <0.05, **P-value <0.01, ***P-value <0.001."

Request 4

Suggestions:

Authors could provide the relative physical location of ADA2 and ADA3 in Arabidopsis thaliana based on the starting and ending base-pair figures in relation to the full length of scaffolds in its genome assembly. This information will be interesting to some researchers because flowering genes are essential for seed-producing plants.

Response 4.

We have provided the physical map of ADA2a, ADA3a and ADA3b in a new supplementary figure 1. And also we provide the AGI code for three genes (ln 329, 214 and 215).